# Cell Cycle Control by Optogenetically Regulated Cell Cycle Inhibitor Protein p21

**DOI:** 10.3390/biology12091194

**Published:** 2023-08-31

**Authors:** Levin Lataster, Hanna Mereth Huber, Christina Böttcher, Stefanie Föller, Ralf Takors, Gerald Radziwill

**Affiliations:** 1Faculty of Biology, Institute of Biology II, University of Freiburg, 79098 Freiburg, Germany; levin.lataster@biologie.uni-freiburg.de (L.L.);; 2Institute of Biochemical Engineering, University of Stuttgart, 70569 Stuttgart, Germany; stefanie.foeller@ibvt.uni-stuttgart.de (S.F.); ralf.takors@ibvt.uni-stuttgart.de (R.T.); 3Signalling Research Centres BIOSS and CIBSS, University of Freiburg, 79098 Freiburg, Germany

**Keywords:** cell cycle, cell cycle inhibitor p21/CDKN1A, G1 arrest, optogenetics, cryptochrome 2, LINuS

## Abstract

**Simple Summary:**

The cell cycle is divided in four phases, the G1 phase for growth in cell size and increased protein biosynthesis, the S phase for the synthesis and replication of DNA, and the G2 phase for preparing the cell for the M phase, the phase of cell division. Cell cycle inhibitors control progression through the cell cycle. The cell cycle inhibitor p21 arrests cells in the G1 phase correlating with a prolonged protein production phase. This effect could be used to increase the production of biotherapeutic proteins. Here, we applied an optogenetic approach to control the function of p21. Optogenetics is an emerging field within synthetic biology and based on genetically encoded light-sensitive elements derived from plants, fungi or bacteria. Optogenetic tools can be used to control biological functions such as signaling pathways, metabolic pathways or gene expression via light with less side effects than when using chemical inducers. In this study, we designed and applied light switches to control the subcellular localization and thereby the function of p21via light. The stimulation of light-regulated p21 increased the number of cells arrested in the G1 phase correlating with the increased expression of a reporter protein. Implementation of this system could be used to optimize the production of biotherapeutic protein.

**Abstract:**

The progression through the cell cycle phases is driven by cyclin-dependent kinases and cyclins as their regulatory subunits. As nuclear protein, the cell cycle inhibitor p21/CDKN1A arrests the cell cycle at the growth phase G1 by inhibiting the activity of cyclin-dependent kinases. The G1 phase correlates with increased cell size and cellular productivity. Here, we applied an optogenetic approach to control the subcellular localization of p21 and its nuclear functions. To generate light-controllable p21, appropriate fusions with the blue light switch cryptochrome 2/CIBN and the AsLOV-based light-inducible nuclear localization signal, LINuS, were used. Both systems, p21-CRY2/CIB1 and p21-LINuS, increased the amounts of cells arrested in the G1 phase correlating with the increased cell-specific productivity of the reporter-protein-secreted alkaline phosphatase. Varying the intervals of blue LED light exposure and the light dose enable the fine-tuning of the systems. Light-controllable p21 implemented in producer cell lines could be applied to steer the uncoupling of cell proliferation and cell cycle arrest at the G1 phase optimizing the production of biotherapeutic proteins.

## 1. Introduction

The cell cycle is a tightly controlled process, in which a multitude of regulatory proteins leads to cell growth and ultimately to mitosis creating two identical daughter cells. It comprises four distinct phases: the growth phase (G1), the synthesis phase of DNA replication (S), the phase of preparing the cell for mitosis (G2) and the mitotic phase (M). The progression through the cell cycle phases is driven by cyclin-dependent kinases (CDKs) and cyclins as their regulatory subunits and is tightly controlled by defined checkpoints [1,2]. The activity of the cyclin-CDK complexes is regulated by members of the CDK-interacting protein/kinase inhibitory protein (CIP/KIP) family and the inhibitor of the kinase (INK) family [3]. Cells arrested in the G1 phase are characterized by an increased cell size, mitochondrial activity and protein biosynthesis [4,5].

The CDK-interacting protein p21 (CDKN1A) is a bifunctional protein. As a nuclear protein, p21 acts as a tumor suppressor and forms complexes with cyclin D-CDK4/6 and cyclin E-CDK2, inhibiting their kinase activity and the progression through the G1/S phase checkpoint [6,7,8,9,10]. Thereby, the cells are arrested in the G1 phase. On the other hand, p21 localized to the cytoplasm functions as an anti-apoptotic protein, also affecting cell motility and proliferation [11]. The subcellular localization of p21 is controlled by the accessibility of a nuclear localization signal (NLS; amino acid residues Arg 140, Lys 141 and Arg 142) [12] and two nuclear export signals (NES1, amino acid residues Leu 75 and Leu 76; NES2, amino acid residues Ile 111, Leu 113, Leu 115 and Leu 117) [13]. The phosphorylation of Thr 145 and Ser 146, neighbored to the NLS, through AKT1 inhibits the translocation of p21 to the nucleus and retains it in the cytoplasm [14]. In addition, the amount of p21 is regulated by a high turnover rate with a half-life time of 20–60 min [8,15,16,17,18,19].

In this study, we aimed to control the function of p21 by regulating its subcellular localization applying an optogenetic approach. Optogenetics allows for the control of biological processes with high spatial and temporal resolution through light, including the recruitment of a protein to a defined compartment, the activity of an enzyme or the assembly of signaling complexes building up signal transduction pathways [20,21,22]. The optogenetic switch AsLOV2 is based on the light–oxygen–voltage-sensitive LOV2 domain of *Avena sativa* phototropin 1 caging the C-terminal helical sequence Jα in the dark and releasing it as unfolded peptide sequence under 450 nm of light exposure [23]. Based on AsLOV, a light-inducible nuclear localization signal (LINuS) was engineered by fusing a NLS to the Jα helix of AsLOV2 [24]. In the dark state, the NLS is caged and not accessible, while upon irradiation with blue light, the NLS is released and the system is translocated into the nucleus in a fast and reversible manner. As a variant of LINuS, the light-inducible nuclear export system (LEXY) enables precise spatiotemporal control over the export of tagged proteins [25].

The second switch used in this study is based on photoreceptor cryptochrome 2 (CRY2) of *Arabidopsis thaliana* and its binding partner the cryptochrome-interacting basic helix–loop–helix protein 1 (CIB1) [26]. CRY2 has two modes of interaction when illuminated with blue light. Either, it forms homo-oligomers or it forms hetero-oligomers by recruiting CIB1 [27,28]. This system has successfully been applied to control plasma membrane recruitment, the function of protein kinases and scaffolding proteins and gene expression [26,29,30].

Here, we used the AsLOV-derived LINuS and the CRY2/CIB1 switch, to control the subcellular localization of mutant versions of p21 and thereby control the function of p21. To simplify the CRY2/CIB1 system, the two components were expressed as a bicistronic construct separated by a P2A “self-cleaving” sequence [31]. Both systems, p21-LINuS and p21-CRY2/CIB1, increased the amounts of cells arrested in the G1 phase correlating with the increased cell-specific productivity of a reporter protein. To our knowledge, this is the first study demonstrating the control of the cell cycle of mammalian cells using light. This system can be transferred to producer cell lines to optimize the production of biotherapeutic proteins.

## 2. Materials and Methods

### 2.1. The Cell Culture and Transfection

Adherent Chinese ovary hamster (CHO)-K1 cells (American Culture Tissue Collection, ATCC, catalog no. CCL 61) and human embryonic kidney (HEK) 293T cells (German Collection of Microorganisms and Cell Cultures, DSMZ, Braunschweig, Germany, catalog no. ACC 635) were cultivated in Dulbecco’s modified Eagle’s medium (DMEM), supplemented with 10% (*v*/*v*) fetal calf serum (FCS) and penicillin (100 U/mL) /streptomycin (100 μg/mL). The cells were cultured in cell culture dishes at 37 °C in a 5% CO_2_ humidified incubator.

### 2.2. Plasmids

All plasmids used in the study were created using Gibson or AQUA cloning [32] with Q5 polymerase (New England Biolabs GmbH, Ipswic, MA, USA). The human p21 cDNA and hamster p21 cDNA inserted into a pcDNA 3.1 backbone used for the CRY2/CIBN system and for the LINuS system, respectively, were ordered from Genescript. The LINuS plasmids were kindly provided by Prof. Barbara DiVentura [24].

### 2.3. Transfection and Western Blot

CHO-K1 cells or HEK293T cells were seeded at 2 × 10^5^ cells/mL in a 6-well plate and transfected with the plasmids indicated using Lipofectamine 3000 (Invitrogen GmbH, Darmstadt, Germany) in accordance with the manufacturer’s protocol. Briefly, 24 h after transfection, the cells were washed once with ice-cold 1× PBS. Subsequently, 200 μL of lysis buffer (20 mM Tris-HCl, pH 7.5; 1 mM EDTA, pH 8; 100 mM NaCl; 0.5% Triton X-100; 0.1% (*w*/*v*) SDS; protease inhibitor (complete protease inhibitor cocktail tablets, Roche Diagnostics GmbH, Mannheim, Germany); 10 mM β-glycerophosphate; 50 mM sodium fluoride; 1 mM sodium orthovanadate; 10 mM sodium pyrophosphate) was added per well and the plate was placed at −20 °C for 15 min. Subsequently, the cell lysate was scraped off, transferred into a microcentrifugation tube, and spun down at 10,000× *g* at 4 °C for 10 min. Before loading onto the gel, they were mixed with SDS loading buffer (2.5% (*v*/*v*) 2-mercaptoethanol; 0.01% (*w*/*v*) bromophenol blue; 10% (*v*/*v*) glycerol; 2% (*w*/*v*) SDS; 62.5 mM Tris-HCl, pH 7.5) and heated at 95 °C for 5 min. If not used immediately, the samples were stored at −20 °C until use. Ten microliters of lysate were loaded onto 10% SDS-PAGE and run at 90 V for 10 min, followed by 120 V for 1 h. Subsequently, a semi-dry protein transfer was performed on a PVDF membrane. The membrane was then blocked for 1 h at room temperature with TBS-T (50 mM Tris-HCl, 150 mM NaCl, pH 7.4, with 0.1% (*v*/*v*) Tween-20) containing 5% BSA. The membrane was incubated with the primary antibody overnight at 4 °C with shaking. On the next day, the membrane was washed three times for 10 min with TBS-T, followed by incubation with the secondary HRP-linked antibody for 1 h at room temperature, and another three washes with TBS-T. The ECL solution supplemented with 0.01% (*v*/*v*) H_2_O_2_ was added to the membrane prior to the detection of chemiluminescence using an ImageQuant LAS-4000 mini system (GE Healthcare, Chicago, IL, USA).

### 2.4. Illumination

Depending on the experimental setup, the adherent CHO-K1 cells and HEK293T cells were cultivated in an appropriate format of cell culture dishes. For cell cycle analyses, the illumination was performed with micro-controller-regulated illumination panels containing LEDs of ~450 nm. For analysis of the SEAP reporter activity, the illumination of the cells was performed with the optoPlate96 [33] ~450 nm LEDs and the illumination protocols was performed with opto-Config-96 [34]. Light-sensitive samples were always handled under dim red or green light.

### 2.5. Flow Cytometry

To determine the distribution of cell cycle phases and the influence of p21 constructs on the cell cycle, flow cytometry was performed. CHO-K1 or HEK239T cells were seeded in a 24-well plate at 0.5 × 10^5^ cells/mL and transfected with the described plasmids using Lipofectamine 3000 (Invitrogen GmbH, Darmstadt, Germany) in accordance with the manufacturer’s protocol. After transfection, all cells were kept in the dark and 24 h later, the cells were illuminated with the specified blue light conditions at 450 nm using self-built light boxes. Subsequent to illumination, CHO-K1 cells were washed once with 1× PBS and detached with Trypsin-EDTA solution before being added dropwise into −20 °C, 100% EtOH while being vortexed to reduce the clumping of the cells. The cells were incubated for at least 24 h at −20 °C, before being spun down and transferred into a 96-well plate. The cells were washed twice with PBS supplemented with 2% BSA and then incubated with 50 µg/mL of propidium iodide (Invitrogen GmbH, Darmstadt, Germany) along with 100 µg/mL of RNase A and 0.1% Triton X100 for 30 min at room temperature. After another washing step with 1× PBS supplemented with 2% BSA, the fluorescence of propidium iodide was measured using Attune Flow Cytometer (Thermo Fisher Scientific, Waltham, MA, USA). The flow cytometry data were analyzed via Kaluza Analysis 2.1 (Beckman Coulter, Brea, CA, USA) using the Micheal H. Fox algorithm.

### 2.6. SEAP Assay

To measure the difference in the protein production of CHO-K1 cells, a SEAP assay was conducted. The cells were seeded in a black 96-well plate with 0.3 × 10^5^ cells/mL and transfected using Lipofectamine 3000 (Invitrogen GmbH, Darmstadt, Germany) in accordance with the manufacturer’s protocol with the corresponding plasmids, including the reporter plasmid for SEAP expression driven by the constitutive EF1α promoter. CHO-K1 cells were kept in the dark for 24 h, before being illuminated under varying blue light conditions for 48 h with 450 nm blue-light LEDs. After illumination, the supernatant was transferred to a 96-well plate and heated at 65 °C for 1 h. The remaining cells were washed with 1× PBS, detached, and counted using a CASY counter (Roche Diagnostics Deutschland GmbH, Mannheim, Germany). The supernatant was then spun down at 1250× *g* for 1 min and mixed with 2× SEAP (20 mM homoarginine, 1 mM MgCl_2_, and 21% diethanolamine; pH 9.8) buffer before the absorbance was measured at 405 nm after the addition of 120 mM pNPP at intervals of 1 min for 1 h. SEAP activity was calculated as described in [35].

### 2.7. Fixation of Cells for Microscopy

For microscopy analysis, cells were seeded on coverslips in 24-well plates. After 24 h, the medium was removed and 200 μL of PFA (4% *v*/*v*, Science Services, E15714) was added to each well and incubated for 15 min at room temperature (RT). After the removal of PFA, the cell-covered coverslips were washed twice with 500 μL of PBS. After the coverslips were briefly dipped in water, they were carefully placed upside down on a drop of Mowiol placed on a clean glass slide. The slides were dried at 37 °C for 60 min and stored in darkness at 4 °C. Fluorescence microscopy was performed using EVOS FL (PEQLAB Biotechnologie GmbH, Erlangen, Germany) microscope (20× objective) with a GFP light cube for eGFP-containing constructs and an RFP light cube for mCherry-containing constructs.

### 2.8. Live Cell Imaging

To visualize the translocation of the p21-LINuS construct, live cell imaging was performed. CHO-K1 cells were seeded in 3.5 cm IBID slides. The transfection of the cells with plasmids containing mCherry for detection was carried out 24 h later using Lipofectamine 3000 (Invitrogen GmbH, Darmstadt, Germany) in accordance with the manufacturer’s protocol. Subsequently, the cells were kept in the dark at 37 °C in a humidified CO_2_ incubator. For microscopy, the cells were placed in a 37 °C humidified CO_2_ chamber (Tokai Hit IncT5, Tokai Hit, Fujinomiya-shi, Japan), and after a 15 min acclimation period, microscopy pictures were taken. The chamber was illuminated using CoolLED-P4000 (CoolLED, London, UK) with a light intensity of 1.33 W/m^2^ at 450 nm. Z-stacks were captured every 5 min in up to 3 selected areas of interest using a 25× objective on an inverted Zeiss LSM880 instrument. The pictures were subsequently analyzed using ImageJ software version 1.53k following [36].

### 2.9. Statistical Analysis

Statistical analysis was performed using Student’s *t*-test or one-way ANOVA combined with Dunnett’s multiple comparisons test. Tests were performed using GraphPad Prism version 9.2.0 for Windows, GraphPad Software, Boston, Massachusetts USA, using * *p* > 0.05, ** *p* > 0.01, *** *p* > 0.001, and **** *p* > 0.0001. The whiskers in the box plots range from the minimum to the lower quartile (the start of the box) and from the upper quartile (the end of the box) to the maximum.

## 3. Results

### 3.1. Light-Controlled p21 Applying a Bicistronic CRY2/CIBN System

The CRY2/CIBN system has been used so far to recruit a cytosolic protein to the plasma membrane, to regulate gene transcription or to reassociate and functionalize split proteins [26]. Here, we used this system to translocate p21 fused to CRY2 (PHR domain, 1–498) into the nucleus via light-dependent binding to CIBN (N-terminal part of CIB1, amino acids 1–100) carrying its own NLS. To promote CRY2/CIBN heteromerization, we used the mutant CRY2low exerting less homo-oligomerization [28]. To prevent interfering by the NLS of p21, the mutant p21NLS^−^ lacking a functional NLS (Arg 140, Lys 141 and Arg 142 replaced by Ala) was used [12]. In the first set of experiments, p21NLS^−^-CRY2low fused to eGFP and CIBN fused to mCherry were coexpressed in HEK293T cells and analyzed via fluorescence microscopy. As expected, p21NLS^−^-CRY2low-eGFP localized to the cytoplasm and CIBN-mCherry localized to the nucleus in the dark. Upon blue light illumination, p21NLS^−^-CRY2low-eGFP translocated to the nucleus and colocalized with CIBN-mCherry demonstrating the functionality of the light-dependent cytoplasm/nuclear translocation system (Appendix A). To express p21NLS^−^-CRY2low and CIBN on one construct, we generated bicistronic constructs, in which both parts were separated by a P2A “self-cleaving” sequence [31] (Figure 1a). In these constructs, CRY2 and CIBN were not equipped with a fluorescent protein. This system worked very efficiently, leading to the expression of p21NLS^−^-CRY2low and CIBN as shown via an anti-HA Western blotting to detect the HA-tagged proteins (Figure 1b and Appendix A).

### 3.2. Functional Characterization of Light-Control p21

To characterize the functionality of the optogenetic approach, the influence of light-regulated p21 on the amounts of cells arrested in the G1 phase and cell-specific productivity with respect to a reporter protein were tested. Therefore, p21NLS^−^ proteins fused to CRY2 or CRY2low and CIBN were co-expressed in HEK293T cells by the respective bicistronic constructs. Briefly, 24 h after transfection, cells were illuminated with intervals of 2 s of blue light (1.33 W/m^2^) and underwent a 3 min dark phase for 48 h (Figure 2a). FACS analyses showed an increase of about 7% in cells arrested in the G1 phase in case the light-regulated p21 was expressed, while blue light illumination only had a low influence on control cells (Figure 2b). Notably, the expression of p21NLS^−^-CRY2/CIBN in HEK293T cells led to an elevated basal level of cells arrested in the G1 phase under dark conditions. Wildtype p21 used as positive control increased the proportion of cells arrested in the G1 phase independently of light exposure to the level obtained via the expression of p21NLS^−^-CRY2low-P2A-CIBN upon blue light exposure (Appendix A). Previous studies have reported that cells in the growth phase G1 have higher ATP levels and are more productive compared to cells in the other phases of the cell cycle [4]. To test cell-specific productivity, a plasmid carrying the reporter protein SEAP under the control of the constitutive EF1α promoter was co-transfected with the bicistronic plasmid encoding p21NLS^−^-CRY2-P2A-CIBN in HEK293T cells. SEAP activity was measured in the supernatant and adjusted to the total cell number to determine cell-specific productivity. The illumination of cells expressing p21NLS^−^-CRY2low-P2A-CIBN increased SEAP activity correlating with the SEAP protein produced per cell by about 60% when compared to that with the dark control. Control cells co-transfected with the vector and the SEAP reporter construct led to no significant effect on cell-specific productivity upon blue light illumination (Figure 2c), whereas the expression of wildtype p21 increased SEAP reporter activity independent of light exposure to a similar level as p21NLS^−^-CRY2low-P2A-CIBN stimulated by blue light did (Appendix A).

#### Optimizing the Light-Control of p21 by the CRY2/CIBN System

So far, we applied one light condition to cells expressing the CRY2/CIBN-based p21 system. To optimize the efficiency of the system, different light conditions were tested, varying the total light dose as well as the intervals of the light and dark phase applied to CHO cells expressing p21NLS^−^-CRY2low-P2A-CIBN for 48 h. An increase of about 40–60% of cell-specific SEAP activity could be reached under the conditions using 30 s, 1 min, and 2 min of blue light and 30 min, 1 h, and 2 h of a dark phase, corresponding to a total light dose of 1 W/m^2^/h, 2 min of blue light and 1 h of darkness corresponding to 2 W/m^2^/h (Figure 3a).

A very low light dose of 2 s of blue light and 1 h of a dark phase as well as a high light dose with 4 min of blue light and 2 h of a dark phase or even constant light for 48 h resulted in no significant increase in cell-specific reporter activity, indicating that defined light/dark settings are necessary for the stimulation of light-regulated p21 in an appropriate way. Increasing the number of repetitions indicated the variance between the experiments resulting from the difference in the transfection efficiency of about 40–60% (Figure 3b). To find the optimal conditions for short light pulses, a set of experiments were performed with 2 s of blue light and different lengths of dark phases. Under the conditions tested, there was a peak in cell-specific SEAP reporter activity using 2 s of blue light and 1 min of darkness (Figure 3c). These data indicate a certain window for the optimal light condition that can be applied in the case of the CRY2/CIBN-based p21 system.

### 3.3. Light-Controlled p21 Applying the LINuS System

As a second optogenetic switch to regulate the cytoplasmic/nuclear localization of p21 and its function as a cell cycle inhibitor, we applied the system LINuS [24]. In this system, the C-terminal NLS is caged in the dark and is released upon exposure to blue light, translocating a protein of interest from the cytoplasm into the nucleus as demonstrated for mCherry (Appendix A). To optimize the light-controlled localization of p21, p21 was fused to LINuS lacking its NLS in addition the 2nd NES (Leu 113 and Leu 115 replaced by Ala). The expression of the fusion constructs was analyzed via Western blotting (Figure 4a,b; Appendix A).

In a first set of experiments, the light-controlled subcellular localization of p21NLS^−^NES^2−^LINuS (in short named p21-LINuS) was determined via live cell microscopy. For live cell microscopy studies, a construct was used carrying mCherry inserted between p21 and LINuS. Cells expressing p21-mCherry-LINuS were illuminated with blue light at an intensity of 1.33 W/m^2^ for 15 min, while z-stack images were captured every 5 min. Upon blue light exposure, the amounts of cells expressing nuclear mCherry increased (Figure 5a). Comparing the fluorescence intensity of mCherry in the nucleus to the cytoplasm at different time points revealed a continuous increase in nuclear mCherry of up to about 20% in light-exposed cells compared to that with the dark control (Figure 5b; Appendix A). To test for the reversibility of the system, cells expressing p21-mCherry-LINuS were first illuminated for 15 min with 1.33 W/m^2^ of blue light, followed by a dark phase of 5 and 10 min and an additional cycle of 5 min of light exposure and dark phases of 5 min and 10 min (Figure 5c). After an increase upon light exposure, the nuclear fluorescence of mCherry decreased in the dark phase, followed by a renewed increase upon the application of the second light stimulus and a decrease in the dark. Thus, the LINuS system allows the reversible nuclear/cytoplasm localization of p21 as monitored here using mCherry.

#### 3.3.1. Light-Controlled G1 Cell Cycle Arrest by p21-LINuS

In the nucleus, p21 functions as an inhibitor to the cell cycle and induces a G1 cell cycle arrest. Therefore, we verified whether or not p21-LINuS is able to induce a G1 arrest in a light-dependent manner. For the following experiments, we used constructs without the insertion of mCherry. Cells expressing p21-LINuS were illuminated with pulsed blue light with 1 s of light exposure (1.33 W/m^2^) and 30 s of a dark phase for 48 h (Figure 6a). FACS analyses showed an increase of about 6% in arrested cells in light-exposed cells compared to that with the dark control (Figure 6b). Cells transfected with the vector alone as a control showed no effect of light exposure on the amounts of cells arrested in the G1 phase. Constructs carrying p21 lacking NLS and the 1st NES behaved similarly to p21NLS^−^NES^2−^LINuS (Appendix A). However, a construct, in which the NLS and both NES of p21 were not functional, led to a light-independent elevated number of cells arrested in the G1 phase. This indicates that at least one of the two NES is necessary for the functionality of the p21-LINuS system.

#### 3.3.2. Light-Controlled Increase in Cell-Specific Productivity with p21-LINuS

Demonstrating the light-dependent increase in nuclear localization and cells arrested in the G1 phase via the optogenetic tool p21-LINuS, we determined the cell-specific productivity known to correlate with a G1 cell cycle arrest. Cells co-transfected with constructs for p21-LINuS and the SEAP reporter construct were illuminated with intervals of 1 s of blue light and underwent 30 s of a dark phase for 48 h as in the experiments before. Under these conditions, cell-specific SEAP activity increased by about 30% in light-treated cells compared to that with the dark control. Cells transfected with the SEAP reporter construct showed no effect on SEAP activity upon light exposure (Figure 6c). Further studies confirmed that constructs lacking one NES of p21 promoted cell-specific SEAP activity, whereas the construct with two non-functional NES prevented the light-inducible regulation of p21 (Appendix A). This is in line with the results shown above that indicate that one NES has to be functional to allow the efficient export of p21-LINuS in the dark state.

## 4. Discussion

In this study, we engineered two optogenetic tools to control the cell cycle using light-regulated p21. While p21-CRY2/CIBN is a two-component system [26] expressed as bicistronic construct in this study, p21-LINuS is a one-component system [24]. According to the known function of nuclear p21 [8], light-activated p21 functions as a cell cycle inhibitor arresting cells in the G1 phase. This correlates with an increase in the cell-specific productivity of 30–60% as monitored here via the activity of the reporter protein SEAP in HEK293T and CHO cells. Both systems function equally well under the right conditions.

For the CRY2/CIBN system, the p21-specific NLS was mutated [12], whereas the two p21-specific NES remain unchanged [13,37]. Thus, the light-induced import of p21-CRY2 depends on its association with CIBN carrying an NLS. In the dark phase, CRY2 and CIBN dissociates and p21-CRY2 is exported by the two p21-specific NES. The LINuS system depends on the uncaging of the C-terminal NLS for nuclear import upon blue light exposure. In the dark phase, the NLS is caged and LINuS is exported by its N-terminal NES. In case of p21NLS^−^-LINuS, an additional mutation of one of the two NES slightly pronounced the effect of blue light on the total amount of cells arrested in the G1 phase; however, the ratio between cells arrested in G1 in the light and the dark state increased significantly. The mutation of both NES in p21NLS^−^-LINuS disturbed the light-independent effects. This can be explained by the inefficient export of p21-LINuS lacking both p21-specific NES in the dark phase. The kinetics determined for p21-LINuS showed efficient import within 15 min of light exposure and reversibility within a further 15 min in the dark phase according to the dynamics reported for the original LINuS system [24]. Via the modulation of NLS and NES, the nuclear/cytoplasmic ratio of p21 as well as the kinetics of the nuclear/cytoplasmic shift can be adjusted and used for the further optimization of the systems.

As shown for the p21-CRY2/CIBN system, an optimal window concerning the total dose of blue light as well as for the interval between the light and dark phase exists. In both systems, p21-LINuS and p21-CRY2/CIBN, short light pulses of 1 s and 2 s followed by a dark phase of 30 s or 60 s, respectively, very efficiently increased cell-specific SEAP activity. It has to be mentioned that the light intensity used in this study (1.33 W/m^2^), did not affect non-transected cells or cells transfected with the vector plasmid or the SEAP reporter plasmid alone under the conditions tested (Figure 2 and Figure 6). However, also in this study, pulse light conditions were superior to constant light. Pulse light may support the dynamics of the shuttling of the p21 fusion proteins between the nucleus and the cytoplasm and thereby promote the productivity of the cells.

## 5. Conclusions

Here, we applied an optogenetic approach to control the function of p21 including cell-specific productivity. Previous studies showed that the expression of wildtype p21 under the inducible control of tetracycline and IPTG increases the cell-specific productivity of a reporter protein or of IgG in a producer cell line [38,39,40]. However, chemical inducers lack spatiotemporal resolution, which is one of the hallmarks of optogenetic approaches [20,41]. Varying the light intervals or the light dose, these systems enable the fast stimulation of the system and the fine-tuning of a target protein, this being p21 in this study, and its function. The implementation of p21-CRY/CIBN or p21-LINuS may allow the generation of producer cell lines for biotherapeutic proteins and the steering of cell proliferation on the one hand and G1 arrest as the production phase of the cells on the other hand by light-controllable p21.

## Figures and Tables

**Figure 1 biology-12-01194-f001:**
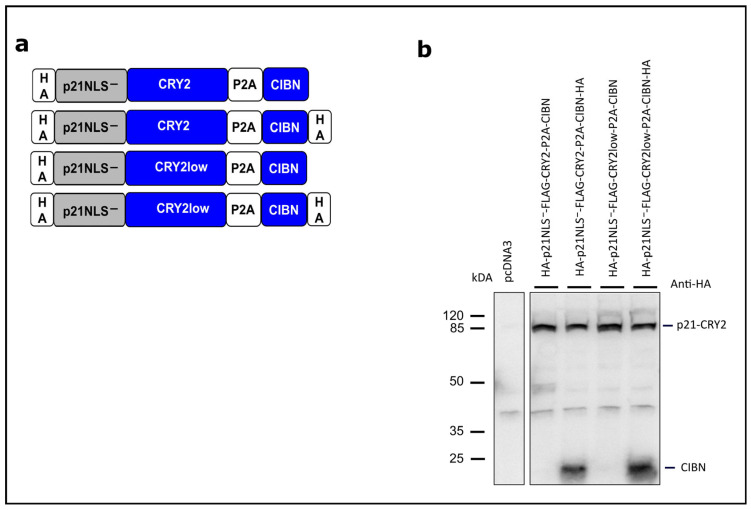
Concept of light-regulated p21 based on the optogenetic switch CRY2/CIBN. (**a**) Schematic drawing of the modular composition of the CRY2/CIBN-based p21. p21/CRY2 fusion proteins and CIBN are expressed as bicistronic constructs separated by a P2A “self-cleaving” sequence. p21NLS^−^: p21 with a mutated, non-functional NLS [12]; CRY2low: a CRY2 mutant with decreased homo-oligomerization efficiency [28]. All constructs carry an additional FLAG tag between p21 and CRY2 not depicted in this scheme. (**b**) Analysis of p21-CRY2-P2A-CIBN constructs. HEK293T cells were transiently transfected with the constructs indicated. 24 h after transfection, the cells were lysed and analyzed via Western blotting with an anti-HA antibody.

**Figure 2 biology-12-01194-f002:**
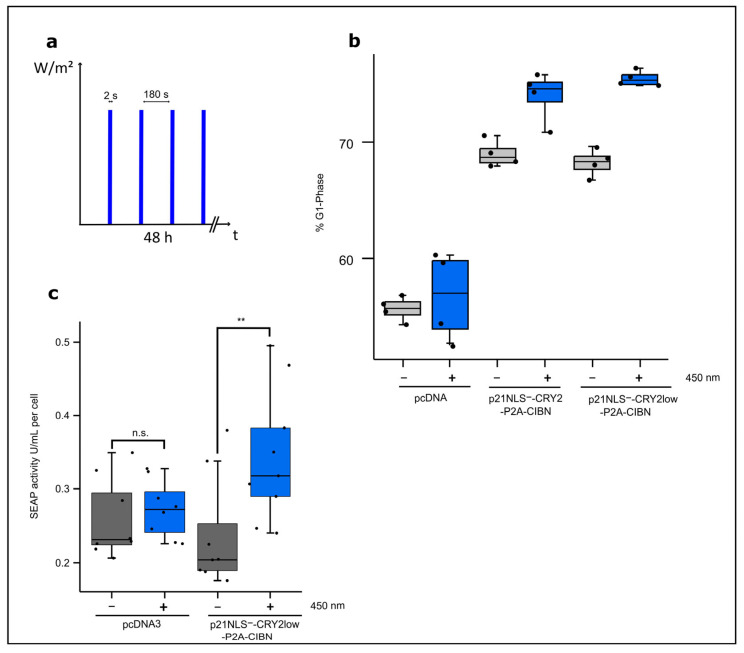
Functional analysis of p21-CRY2/CIBN. (**a**) Illumination scheme. (**b**) Percentile of cells arrested in the G1 phase after blue light illumination. HEK293T cells were transfected with either p21NLS^−^-CRY2-P2A-CIBN, p21NLS^−^-CRY2low-P2A-CIBN or the vector pcDNA3 as a negative control. Briefly, 24 h after transfection, cells were illuminated with 2 s intervals blue light (1.33 W/m^2^) and 3 min of darkness for 48 h. Afterwards, the cells were fixed with ethanol and treated with propidium iodide to determine the distribution of the cell cycle phases via FACS analysis. Cell cycle distribution was quantified with the help of the Michel H. Fox algorithm. Every dot represents one well of a 24-well plate (n = 4). The black boxes represent cells that were kept in the dark, while the blue boxes represent the cells illuminated with blue light with the aforementioned conditions. (**c**) The light-induced increase in the cell-specific reporter’s SEAP activity via the expression of p21-CRY2/CIBN. HEK293T cells were co-transfected with p21NLS^−^-CRY2low-P2A-CIBN and the reporter plasmid expressing EF1-driven SEAP, while the controls were only transfected with the reporter plasmid. Briefly, 24 h after transfection, the cells were illuminated with 2 s intervals blue light (1.33 W/m^2^) and underwent 3 min of a dark phase for 48 h. SEAP activity was measured in the cell supernatant and adjusted to the cell count to determine the SEAP activity per cell. The dots represent one well in a 96-well plate (n = 9). Statistical analysis was performed with the Student’s *t*-test using ** *p* > 0.01; n.s.: non-significant.

**Figure 3 biology-12-01194-f003:**
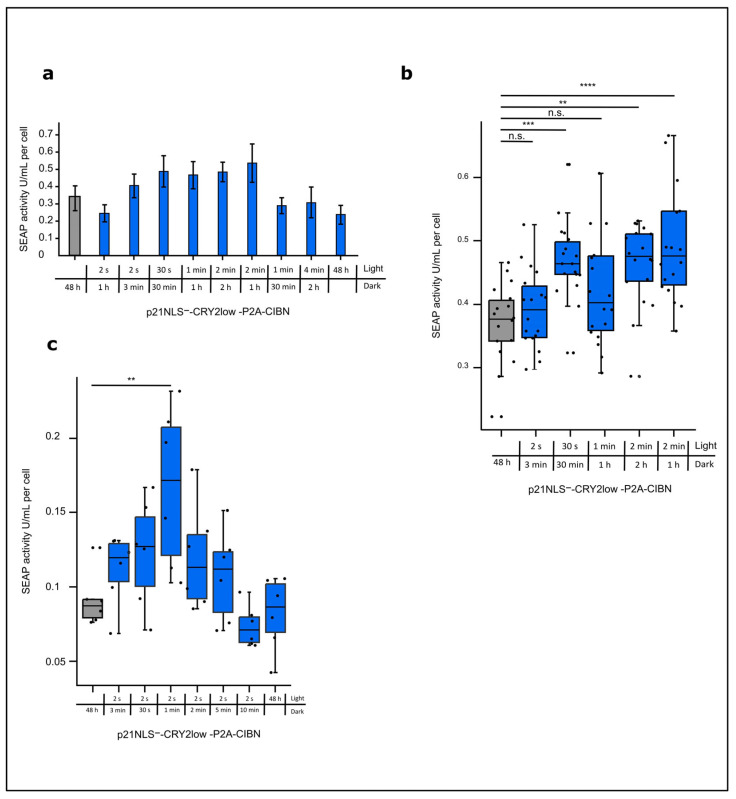
Light-induced increase in cell-specific SEAP reporter activity via the expression of p21-CRY2/CIBN depending on light intensity and pulse setting. (**a**) CHO cells were co-transfected with p21NLS^−^-CRY2low-P2A-CIBN and the SEAP reporter plasmid. Briefly, 24 h after transfection, the cells were treated with different total light amounts with varying light pulses as indicated over 48 h. The cell supernatant was collected and SEAP activity was measured and adjusted to the cell count (see Appendix A), to determine the SEAP activity per cell (n = 6). (**b**) For selected conditions, the experimental setup was studied more intensively. The dots represent one well in a 96-well plate (n = 18). (**c**) CHO cells were co-transfected with p21NLS^−^-CRY2low-P2A-CIBN and the SEAP reporter plasmid. Briefly, 24 h after transfection, the cells were treated with intervals of blue light pulses (1.33 W/m^2^) lasting 2 s and various dark phases as indicated for 48 h. The SEAP activity was adjusted to the cell count to determine the SEAP activity per cell (n = 6). Statistical analysis was performed using ANOVA combined with Dunnett’s multiple comparisons test using ** *p* > 0.01, *** *p* > 0.001, and **** *p* > 0.0001; n.s.: non-significant.

**Figure 4 biology-12-01194-f004:**
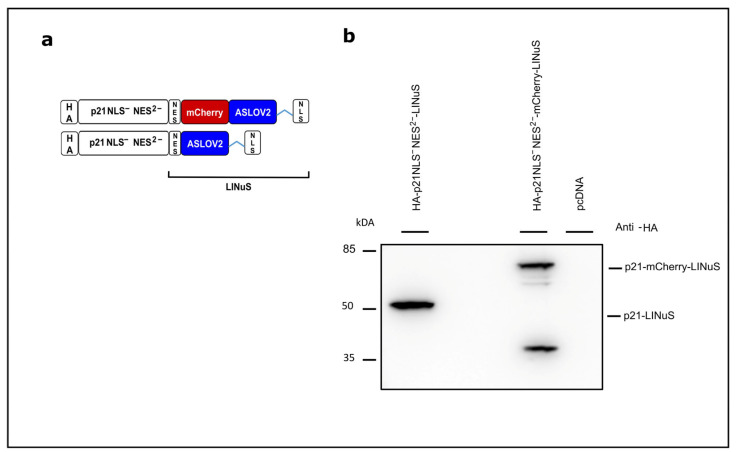
Concept of light-regulated p21 based on the LINuS system. (**a**) Overview of the LINuS-based p21 constructs. p21NLS^−^NES^2−^: p21 with mutated non-functional NLS and non-functional 2nd NES; mCherry: red fluorescent protein. Both constructs are equipped with an N-terminal HA-tag. (**b**) Expression of the p21-LINuS constructs in CHO cells. Cells were transiently transfected with the constructs indicated. Briefly, 24 h after transfection, the cells were lysed and analyzed via Western blotting with an anti-HA antibody.

**Figure 5 biology-12-01194-f005:**
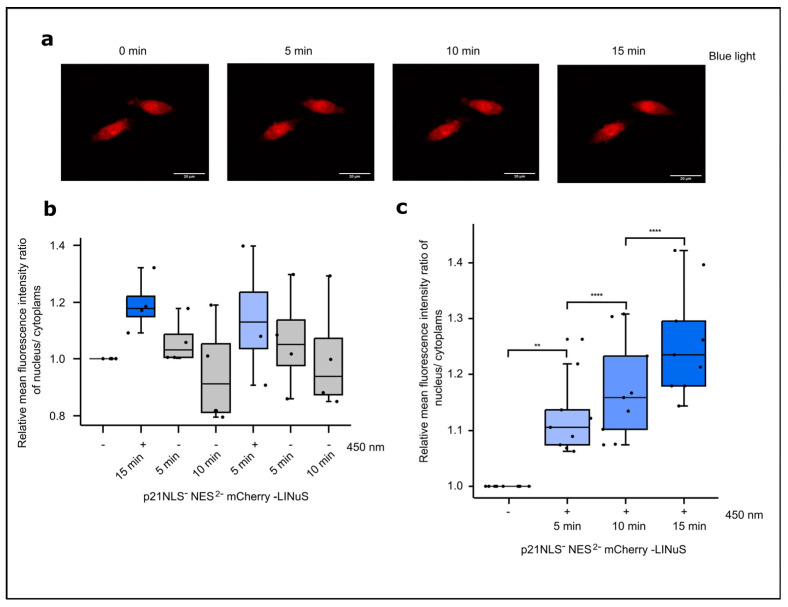
Light-induced translocation of p21-LINuS from the cytoplasm into the nucleus. (**a**) Live cell imaging of CHO cells expressing p21NLS^−^-NES^2−^-LINuS for 24 h and subsequently kept in the dark or illuminated with blue light (1.33 W/m^2^) for 5, 10 or 15 min. (**b**) Quantification of the ratio of the relative mean fluorescence intensity of cytoplasmic and nuclear mCherry in CHO cells expressing p21NLS^−^-NES^2−^-mCherry-LINuS after 5 min of blue light illumination with dark pauses as indicated. Images were evaluated at the indicated time points. Each dot represents one cell (n = 4) and the ratios were normalized to the dark state of the cell. (**c**) The quantification of the ratio of the relative mean fluorescence intensity of cytoplasmic and nuclear mCherry in CHO cells expressing p21NLS^−^-NES^2−^-mCherry-LINuS after blue light illumination for three time intervals as indicated. Each dot represents one cell (n = 9) and the ratios were normalized to the dark state of the cell. Statistical analysis was performed using ANOVA combined with Dunnett’s multiple comparisons test using ** *p* > 0.01, and **** *p* > 0.0001.

**Figure 6 biology-12-01194-f006:**
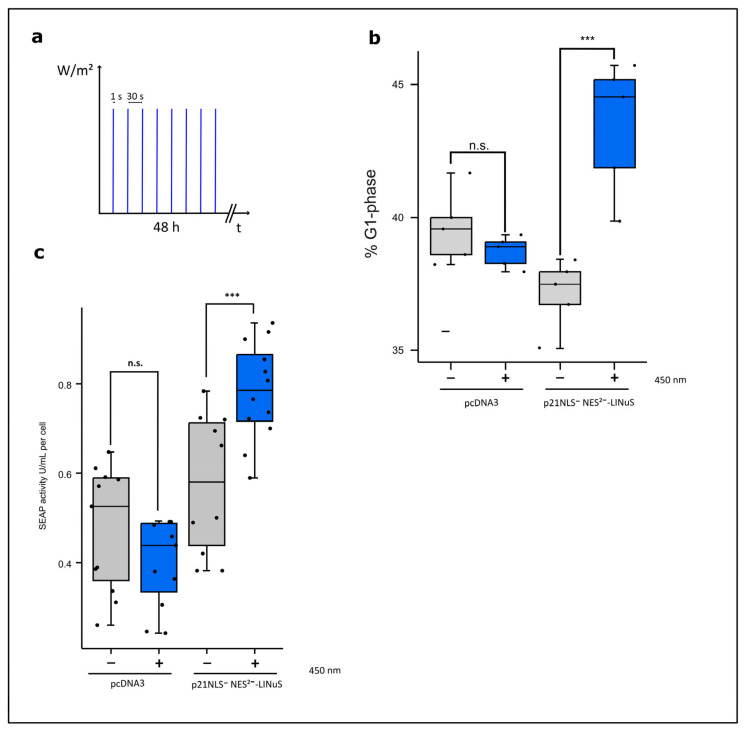
Light-controlled G1 cell cycle arrest and cell-specific productivity induced by p21-LINuS. (**a**) Illumination scheme. The cells were illuminated with intervals of 1 s of blue light and 30 s of a dark phase for 48 h. (**b**) The percentile of cells arrested in the G1 phase after blue light illumination. CHO cells were transfected with p21NLS^−^-NES2^−^-mCherry-LINuS. Briefly, 24 h after transfection, the cells were illuminated with intervals of 1 s of blue light (1.33 W/m^2^) and underwent a 30 s dark phase for 48 h. Afterwards, the cells were fixed with ethanol and treated with propidium iodide to determine the distribution of the cell cycle phases with and without blue light illumination via FACS analysis. Cell cycle distribution was determined with the help of the Michel H. Fox algorithm. Every dot represents one well of a 24-well plate (n = 5). (**c**) Light-induced cell-specific SEAP reporter activity determined via the expression of p21-LINuS. Cells were transfected with the plasmid p21NLS^−^-NES2^−^-mCherry-LINuS and the SEAP reporter plasmid or the SEAP reporter plasmid alone as a control. Briefly, 24 h after transfection, the cells were illuminated with intervals of 1 s of blue light (1.33 W/m^2^) and underwent a 30 s dark phase for 48 h. The dots represent one well in a 96-well plate (n = 11). Statistical analysis was performed with Student´s *t*-test using *** *p* > 0.001; n.s.: non-significant.

## Data Availability

The data presented in this study are available on reasonable request from the corresponding author.

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
