# Peer review of "Cell Cycle Control by Optogenetically Regulated Cell Cycle Inhibitor Protein p21"

_biology, 2023, doi:10.3390/biology12091194_

Round 1

Reviewer 1 Report

In this study, the authors have developed two tools to manipulate the CDK inhibitor p21 with blue light and investigated their effects on the cell cycle: first, a light-dependent CRY-CIB heterodimerization system for p21 nuclear translocation; second, a light-dependent NLS LINuS-based nuclear translocation system for p21. In both systems, blue light-induced cell cycle arrest and increased protein synthesis have been observed.

Optical manipulation of the cell cycle is fascinating and I commend the authors for their attempt. However, some of the points this manuscript argues are not sufficiently validated by experiments and analyses. I provide some comments below that would be worth addressing in a revised version.

Major comments

1. The authors did not show any experimental evidence that p21NLS--CRY2 is translocated to the nucleus upon blue light illumination, thereby inducing cell cycle arrest. 

2. There is no positive control for the effect of p21 on cell cycle arrest. For example, how much p21 overexpression increases the proportion of cells in G1-phase and SEAP activity. 

3. I am not entirely sure why the authors evaluated cell-specific productivity as a marker of G1 arrest. For example, Figure 2b shows that expression of p21NLS-CRY2-P2A-CIBN or p21NLS-CRY2low-P2A-CIBN alone in the dark causes G1 arrest compared to pcDNA control cells, whereas Figure 2c shows no effect of expression of the same proteins on SEAP activity in the dark between control and p21NLS-CRY2-P2A-CIBN expressing cells. 

4. In line 272, the authors argue that “The increase in the cell-specific productivity correlated with a decreased total cell count indicating that light-controlled p21 affect the cell proliferation by increasing the number of arrested cells (Figure S1)”. But, I could not find any correlation between Figure 3a and Figure S1. 

5. In line 270, “Statistical analysis was performed with the Student´s t-Test using * p>0,05, ** p>0,001, ***p>0,0001”. These data should be analyzed by statistically robust methods such as multiple comparison test, but not the Student’s t-test. 

6. In Figure 5, p21-mCherry-LINuS is substantially localized at the nucleus before blue light stimulation, although the authors quantified only relative increase in translocation of p21-mCherry-LINuS upon blue light illumination. Again, there are no positive and negative controls for the quantification, such as mCherr-NLS and mCherry-NES. 

Minor comments

7. It may be better to include a schematic diagram showing light-induced nuclear translocation of p21-CRY2 through the binding to CIBN at the nucleus.

8. In Figure 3c, the dark 48-hour data have no whiskers on the top. In addition, there is no information about the whiskers in box-whisker plots. 

9. In several graphs, the Y-axis values should use a period but not a comma.

Reviewer 2 Report

The authors in this work created an elegant optogenetically regulated systems in order to control p21 protein and increase the number of cells in G1 phase of the cell cycle, which has potential applications in both biotechnology and biopharmaceutical applications.

Conclusions are, in general, supported by the data, however some modifications should be made.

In Fig. 3 b and c ANOVA is a more appropriate statistical test as authors are basically comparing more then two groups and if the comparison is performed by t-test it is more likely that the result would turn out significant, so the data should be assessed by ANOVA. The same is true for data in Fig. 5 b and c. Also, in Fig. 5 there is no indication as to what statistical tests were used and how the data was assessed (my assumption is that also t-test was used).

Presentation of data would benefit from separating schematic panels of figures into separate figures or rearranging panels in different way to increase clarity.

Sentence in lines 274-276: the observation should be discussed in more detail in the discussion section.

line 273: affect should be affects

line 332: compared the dark should write compared to the dark

line 333: for should be substituted with as

Proofread for minor mistakes.

Round 2

Reviewer 1 Report

Overall, the authors have done a mostly adequate job of responding to the reviewers' comments.